# Seeing through the mess: evolutionary dynamics of lexical polysemy

**Andreas Baumann[1], Andreas Stephan[2,3], Benjamin Roth[1,2]**

[1]Faculty of Philological and Cultural Studies, [2]Faculty of Computer Science,
[3]UniVie Doctoral School Computer Science DoCS
University of Vienna, Austria
{andreas.baumann,andreas.stephan,benjamin.roth}@univie.ac.at

## Abstract

Evidently, words can have multiple senses. For example, the word *mess* refers to a place to have food or to a confusing situation. How exactly multiple senses emerge is less clear. In this work, we propose and analyze a mathematical model of the evolution of lexical meaning to investigate mechanisms leading to polysemy.

This model features factors that have been discussed to impact the semantic processing and transmission of words: word frequency, non-conformism, and semantic discriminability. We formally derive conditions under which a sense of a word tends to diversify itself into multiple senses that coexist stably.

The model predicts that diversification is promoted by low frequency, a strong bias for non-conformist usage, and high semantic discriminability. We statistically validate these predictions with historical language data covering semantic developments of a set of English words. Multiple alternative measures are used to operationalize each variable involved, and we confirm the predicted tendencies for twelve combinations of measures.

## 1 Introduction

In natural language, lexical polysemy, i.e., the presence of multiple senses for a single word form, is the rule rather than the exception. The word *mess*, for instance, can denote, among other things, a room in which food is served, semi-liquid food, a confusing situation, or a physical state of disorder. From a communicative point of view, the fact that one form refers to multiple senses is, at first sight, sub-optimal given that ambiguity acts against successful communication. Yet, populations of speakers sustain a multitude of polysemous words in their communicative systems, *viz.*, languages.

Where does this semantic diversity come from? Evidently, multiple senses of a word do not just simply appear. Rather, word meaning evolves over

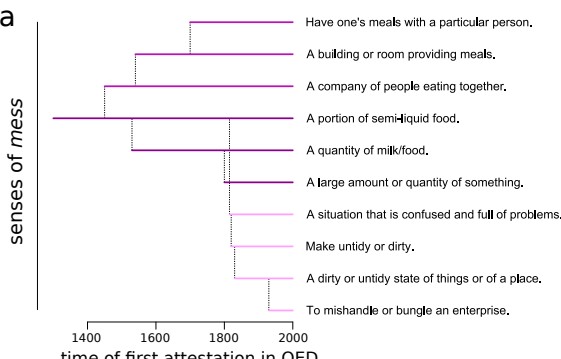

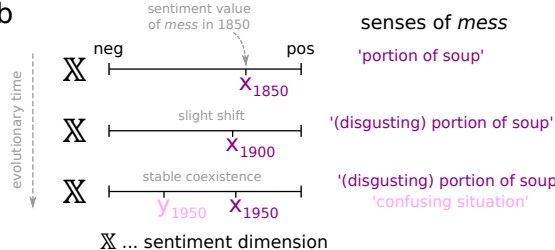

Figure 1: (a) Evolutionary tree of the senses of *mess* based on first dates of attestation as documented in the Oxford English Dictionary (OED). (b) Schematic sketch of the diversification of *mess* in the semantic dimension of sentiment (negative to positive).

time and diversifies itself as additional senses are established in the population of speakers (Traugott, 1985; Deane, 1988; Sagi et al., 2011; Mitra et al., 2014; Hamilton et al., 2016; Hu et al., 2019; Schlechtweg et al., 2020). Figure 1a displays the evolutionary tree of the semantics of the word *mess*.

The goal of this paper is to identify conditions under which semantic diversity is enforced with the help of analytic models of the population dynamics of words. More specifically, we define a model of the spread of words through a population of speakers in which word transmission is governed by social and cognitive factors that have been suggested to be relevant to semantic change: word frequency, non-conformism, and semantic discriminability. An analytical assessment of our model shows that these factors indeed affect the

*tendency* to establish additional senses in speaker populations. The predictions of the model are subsequently tested against empirical language data.[1]

## 2  Approach

We consider three factors that have been suggested to be relevant to lexical transmission, semantic processing, and change (frequency, non-conformism, discriminability; see Background section 3 below). The question is if and how these factors impact the diversification of a word's meaning, and, more importantly, what the potential mechanisms leading to diversification might be. Our approach to tackle these questions involves two parts, namely a theoretical and an empirical part.

On the theoretical level, we model the evolution of a word's meaning over time and analyze under which conditions a word's meaning is split into two separate and stably coexisting senses. This model captures the population dynamics of the semantic variants of a word, that is, we investigate how word variants spread through a population of speakers. Interactions among speakers and how they transmit word variants are governed by effects of frequency, non-conformism, and discriminability that we assume beforehand.

More concretely, we assume (i) that the successful transmission of a word variant from one individual to another is more likely if it is used frequently, (ii) that rare linguistic behavior can have an advantage during interactions as it stands out, and (iii) that the cognitive ability to discriminate between senses impacts what is actually perceived as the same or different linguistic behavior in interactions. Table 1 shows more details about these assumptions. Note, crucially, that none of these assumptions is about semantic *long-term* evolution *per se*. All of them are exclusively about how individuals communicate and process language in *short-term* interactions.

Based on the model, we deduce the evolutionary dynamics of a word's meaning and we derive which effects of frequency, non-conformism, and discriminability on the tendency to diversify meaning the model would predict.

Mathematically, our model consists of two stacked dynamical systems defined by ordinary differential equations: one that defines the long-term evolutionary dynamics of meaning, and one

that defines the short-term dynamics of lexical usage in a speaker population. In combination, the model lets us show that diversification of meaning is encouraged for words subject to low frequency, high non-conformism biases, and high semantic discriminability (cf. Table 1).

Note that although the model would, provided all necessary historical information, *theoretically* allow us to obtain semantic trees for single words and single semantic dimensions (like 'sentiment', 'edibility', 'objectness') similar to the one in Figure 1a, this is *not* what we do. We also do not need to focus on the linguistic behavior of individual speakers. Rather, the strength of our model is that we can use it to predict more generally under which conditions semantic diversification can occur on the population level. Thus, our model reveals potential mechanisms governing the establishment of multiple coexisting senses.

On the empirical level, we measure the impact of frequency, non-conformism, and discriminability on the tendency to diversify meaning. We base our analysis on historical language data covering about 3000 English words (Hu et al., 2019). Based on this data set, we show that the predictions of the theoretical model are also supported on empirical grounds.

In what follows, we motivate our choice of factors in semantic evolution, present an outline of the evolutionary model, its underlying population dynamics, and its predictions of how long-term evolution unfolds. After that, we describe the setup and outcome of our empirical account. The discussion section links both parts of this study.

## 3  Background

We adopt the established definition of polysemy as a one-to-many form-meaning mapping that comes about via changes on the meaning side. The key criterion here is historical relatedness (Leech, 1974; McMahon, 1994). This contrasts with the phenomenon of homophony (e.g., *steak* vs. *stake*), which results from changes on the form side (see Valera, 2020, for a more recent discussion).

Multiple factors have been suggested to impact semantic change and the evolution of polysemy. In this research, we consider three of them that have been discussed in the linguistic, cognitive, and computational literature.

The most prominent factor that was suggested to impact lexical semantics is the word **frequency**. In

---

[1]Code and data available at https://gitlab.com/andreas.baumann/evolution_lexical_polysemy

| | Frequency $\varphi$ | Non-conformism bias $\beta$ | Discriminability $\delta$ |
|---|---|---|---|
| Definition | Number of occurrences of a word variant | Tendency at which a word variant is used in non-conformist vs. conformist settings | Ability to distinguish between two semantic uses of a word |
| Empirical measurement | Normalized token frequency of a word (contemporary vs. historical) | Fraction of frequency in non-conformist genres vs. frequency in conformist genres | Lexical concreteness ratings and average correlation between sentence embeddings of all sense descriptions of a word |
| Model assumptions about interactions | High exposure increases chance of successful adoption in interactions | Imposes advantage in the propagation of the word variant when rare (common strategies lose their distinctive status) | Modulates effects of (non-) conformist behavior in interactions among individuals using different variants of a word |
| Prediction | $T \sim 1/\varphi$ ✓ | $T \sim \beta$ ✓ | $T \sim \delta$ ✓ |

Table 1: Overview of the main model parameters, how they are defined and measured, and their respective impact on the tendency to diversify meaning $T$ as predicted by the model. A check-mark (✓) indicates whether the predictions are empirically supported.

their seminal contribution, Hamilton et al. (2016) have demonstrated that high-frequency words are, on the historical time scale, more stable than low-frequency words (but see Dubossarsky et al. (2017) for criticism).

The negative relationship between semantic change and frequency was suggested to be an effect of entrenchment (Hamilton et al., 2016; Cassani et al., 2021; Baumann et al., 2023). Frequent words are typically learned early, and this leads to cognitive entrenchment, i.e., routinization of their processing and usage (Bybee, 2006). By consequence, it is less likely for high-frequency words to change their semantic profile than this might be the case for low-frequency words that could be, perhaps erroneously, used in novel contexts.

The second factor that was suggested to increase diversity in the domain of cultural evolution is that of **non-conformist** behavior, i.e., to behave differently than the majority does. Non-conformist behavior entails what is referred to as negative frequency dependence (Efferson et al., 2008; Boyd and Richerson, 1988; Doebeli and Ispolatov, 2010a). That is, behavioral strategies are particularly successful if they are rarely employed.

Non-conformist behavior is present in the linguistic domain as well. For one, several phenomena of linguistic innovations have been discussed under the notion of linguistic extravagance. Speakers have been argued to choose unorthodox linguistic strategies to stand out, get recognized, and in turn to promote the transmission of their message (Ungerer and Hartmann, 2020; Detges and Waltereit, 2002; Petré, 2017; Hein, 2017; Haspelmath, 2000). Often, extravagant linguistic behavior extends to the semantic domain. So, originally negative words like *awesome* have gained a pos-

itive connotation (this particularly holds true for slang with its function to signal out-group membership) (Fajardo, 2019). The initial salience of these words, however, becomes weaker the more abundantly they are used.

Non-conformism is counteracted by conformism, i.e., the disproportionally high tendency to behave just like the majority does (Henrich and Boyd, 2002). Conformism functions as stabilizing mechanism in the transmission of communicative systems. Intuitively, conformism biases should prevent semantic ambiguity since the introduction of novel senses is discouraged.

The third factor, **discriminability**, refers to an individual's ability to discriminate between two contextual uses of a word (Miller and Charles, 1991; Miller, 1999). Intuitively, if discriminability is high then individuals are less likely to view two different uses of a word as belonging to the same sense (for instance, if individuals can discriminate well, they would suggest the word *mess* to have two different senses in the utterances *I see through the mess* and *I have a burger in the mess*). There is much less research on the effect of semantic discriminability on language change than for the other two factors discussed above, but modeling and simulation results suggest that high discriminability entails more complex signaling inventories (Chaabouni et al., 2021; Imel, 2023). From a cognitive point of view, interdependencies among word utterances are weaker if they are (perceived as being) different from each other, i.e., in the case of high discriminability (den Heyer and Briand, 1986). This also extends to interdependencies imposed by (non-)conformism pressures in interactions among individuals (Kim and Hommel, 2015).

## 4 Theoretical analysis

Our model has the following components: We assume that **words** are transmitted through populations of **speakers**. At any point in time, each word has a set of one or more **senses** that we take to constitute a word's **meaning**. We assume that each word sense can be represented as real value on a **semantic dimension**.

More specifically, we assume that $x$ takes a value on a continuous scale so that $x \in \mathbb{X}$ where $\mathbb{X} \subseteq \mathbb{R}$ is a real-valued interval representing the range of possible values on the semantic dimension. For instance, $\mathbb{X}$ could represent the word's sentiment ranging from 0 (negative) to 1 (positive). Then, a sense of a word like *mess* might have had a rather positive sentiment of about $x = 0.8$ in the middle of the 19th century.

Speakers might introduce a new **variant** of an already existing word sense. This happens, for instance, when someone starts using a word sense in a slightly different context (e.g., 'potentially disgusting soup' rather than just 'soup'). When this innovative behavior is adopted by the speaker population, the semantic value $x$ of a word sense changes within $\mathbb{X}$.

Note that this only means that a certain semantic property of a sense changes. The introduction of an innovative variant of an already existing variant does not automatically entail that both of them coexist stably in the speaker population. Stable coexistence would mean that the semantic value representing one sense splits into two variants defined by two values $x_1$ and $x_2$ within $\mathbb{X}$ that now represent two separate senses (i.e., a more positive and a more negative sense of *mess*) that are both sustained by the speaker population. Such a scenario is schematically visualized in Figure 1b. The crucial question is under which conditions are splits like this possible?

To shed light on this question, we use the *canonical equation of adaptive dynamics* which models the evolution of continuous properties in populations of individuals (Metz et al., 1995; Dieckmann and Metz, 2006; Dercole and Rinaldi, 2008; Doebeli, 2011). Theoretically, the equation was shown to be linked to other well-studied models of evolution such as replicator dynamics (from evolutionary game theory) or the Price equation (Page and Nowak, 2002; Meszena et al., 2002).[2]

In our context, the canonical equation of adaptive dynamics models change in a semantic property. More specifically, it defines the rate of change of $x$ in the semantic subspace $\mathbb{X}$ as a function of (a) the number of users of the word variant characterized by $x$, and (b) the fitness of the word variant relative to another variant:

$$\dot{x} = C \cdot \hat{U}(x) \cdot \left. \frac{\partial f(x,y)}{\partial y} \right|_{y=x} \qquad (1)$$

Here, $C > 0$ is a constant defined by the rate and variance of semantic innovations that is independent of the semantic property $x$. Furthermore, $\hat{U}(x)$ denotes the number of users $U$ of the word sense characterized by the value $x$ on the semantic dimension at population dynamic equilibrium.[3] Intuitively, this means that a higher number of users facilitate larger amounts of change (because each user could introduce an innovation). Finally, $f(x,y)$ denotes the fitness, i.e., the growth rate of a word variant with a slightly different value of the semantic property $y \in \mathbb{X}$ in a population of individuals mainly using the word variant characterized by value $x$. This quantity is referred to as *invasion fitness* because it determines whether or not a new variant can successfully invade to be used by the speaker population (for a visualization of $f$, see Figure 4 in Appendix A.1.2). The term $\partial f(x,y)/\partial y|_{y=x}$ determines the direction of evolution.

Equation (1) can exhibit different types of long-term behavior (Metz et al., 1995). Two relevant scenarios are (a) convergence to a steady-state, i.e., the existence of an *attracting evolutionarily stable strategy* in $\mathbb{X}$ and (b) diversification, i.e., the existence of an *evolutionary branching point* in $\mathbb{X}$ (Geritz et al., 1997). The latter case implies the stable coexistence of two different variants (formally, two stable equilibria of equation (1)). In the following, we investigate conditions for the existence of a branching point, as this relates to polysemy.

Whether or not such points exist in $\mathbb{X}$ for equation (1) depends on the quantities $\hat{U}(x)$ and $f(x,y)$. Both of them are determined by the underlying population dynamics of word variants. The population dynamics of a single word variant characterized

---

[2]This equation has found applications in biology to study the evolution of phenotypic traits like drug resistance in infectious diseases (Dieckmann et al., 2002), in economics to examine technological change leading to efficiency gains (Dercole et al., 2008), or in cultural studies to model the evolution of religious behavior (Doebeli and Ispolatov, 2010b).

[3]This equilibrium is reached when the variant has successfully spread through the speaker population, i.e., when its number of users stops growing. See Appendix A.1.1.

by value $x$ are defined by an ordinary differential equation (ODE) tracing the change in the number of users $U(x)$ of that variant. The change in the number of users is influenced by the factors outlined in Section 3. The dynamics are based on the following **assumptions**:

(I) **Population structure:** Individuals either use or do not use a word variant. We assume homogeneous mixing of users $U$ and individuals not using the word variant, i.e., non-users $N$.

(II) **Transmission:** Whenever a user and a non-user meet, successful transmission of a word variant depends on word *frequency $\varphi$* and an adoption rate $\alpha$. Adoption rate $\alpha$ is assumed to be a smooth function of $x$. We assume that there exists a value $x_0$ within $\mathbb{X}$ maximizing $\alpha$ (arguably, adoption is not independent from semantic properties).

(III) **Negative frequency dependence:** Whenever a user meets another user of the same word variant they abandon it at a rate $\beta$, the *non-conformism bias*. This bias is reduced by a propensity to behave in a conformist way. When a user meets another user with a slightly different variant, non-conformism biases apply as well but they are reduced by the perceived distance between the two variants. This distance depends on *discriminability $\delta$*.

The ODE modeling the change in the number of users $U$ given assumptions (I-III) reads

$$\dot{U} = \varphi\,\alpha(x)UN + \kappa(\Delta = 0)UU - \nu(\Delta = 0)UU, \tag{2}$$

with a conformism rate $\kappa(\Delta) = \kappa_0 \exp(-1/2 \cdot \Delta^2\delta^2)$, a non-conformism rate $\nu(\Delta) = \nu_0 \exp(-1/2 \cdot \Delta^2\delta^2)$, and $\nu_0 - \kappa_0 =: \beta$. Here, $\Delta = x - y$ denotes the difference between two semantic property values $x$ and $y$. A detailed description of this ODE can be found in Appendix A.1.1.

This leads us to the following

**Proposition 1.** *Consider a semantic dimension $\mathbb{X}$. Assume that innovations in $\mathbb{X}$ occur in relatively small steps and that new innovations occur only after a previous innovation has either spread successfully or disappeared.*

*Given the population dynamics of single word variants characterized by a single value in $\mathbb{X}$ as defined in (2), the canonical equation of adaptive dynamics in (1) shows an evolutionary branching point if*

*(i) frequency $\varphi$ is sufficiently low,*

*(ii) non-conformism bias $\beta$ is sufficiently high,*

*(iii) and discriminability $\delta$ is sufficiently high.*

A proof is shown in Appendix A.1. In short, the model predicts the tendency to diversify meaning to be (i) negatively associated with frequency, (ii) positively associated with non-conformism biases and (iii) positively associated with discriminability.

## 5 Empirical analysis

We test if predictions (i-iii) resulting from our theoretical analysis are supported empirically. To do so, we investigate a sample of words and check if (i) frequency, (ii) non-conformism biases, and (iii) discriminability are associated with the tendency to diversify meaning in this sample. That is, our model would predict words with high non-conformism biases, high discriminability between senses, and low frequency to become more polysemous over time.

We base our analysis on a set of 3358 words compiled by Hu et al. (2019) containing information about the semantic development of words. In this data set, each word is accompanied by (i) a list of the descriptions of all of its senses[4] taken from the Oxford English Dictionary (OED), and (ii) for each decade in the 19th and 20th century, the estimated probabilities for all senses of that word, originally derived from the Corpus of Historical American English (COHA) (Davies, 2010) through SentenceBERT embeddings. The development of the distribution of senses of the word *mess* is shown in Figure 2a.

After only considering words with subjective concreteness ratings in Brysbaert et al. (2014) (see 5.3 below), we ended up with a set of 3165 words. We restrict our analysis to all decades between 1850 and 2000 since earlier subcorpora of COHA are relatively small and display a different distribution of genres.[5]

---

[4]Note, also, that the observation period does not cover major sound shifts in the history of English that might have resulted in substantial increase of homophones.

[5]Conducting the analysis with all decades from 1800 to 2000 does not substantially change the results, though.

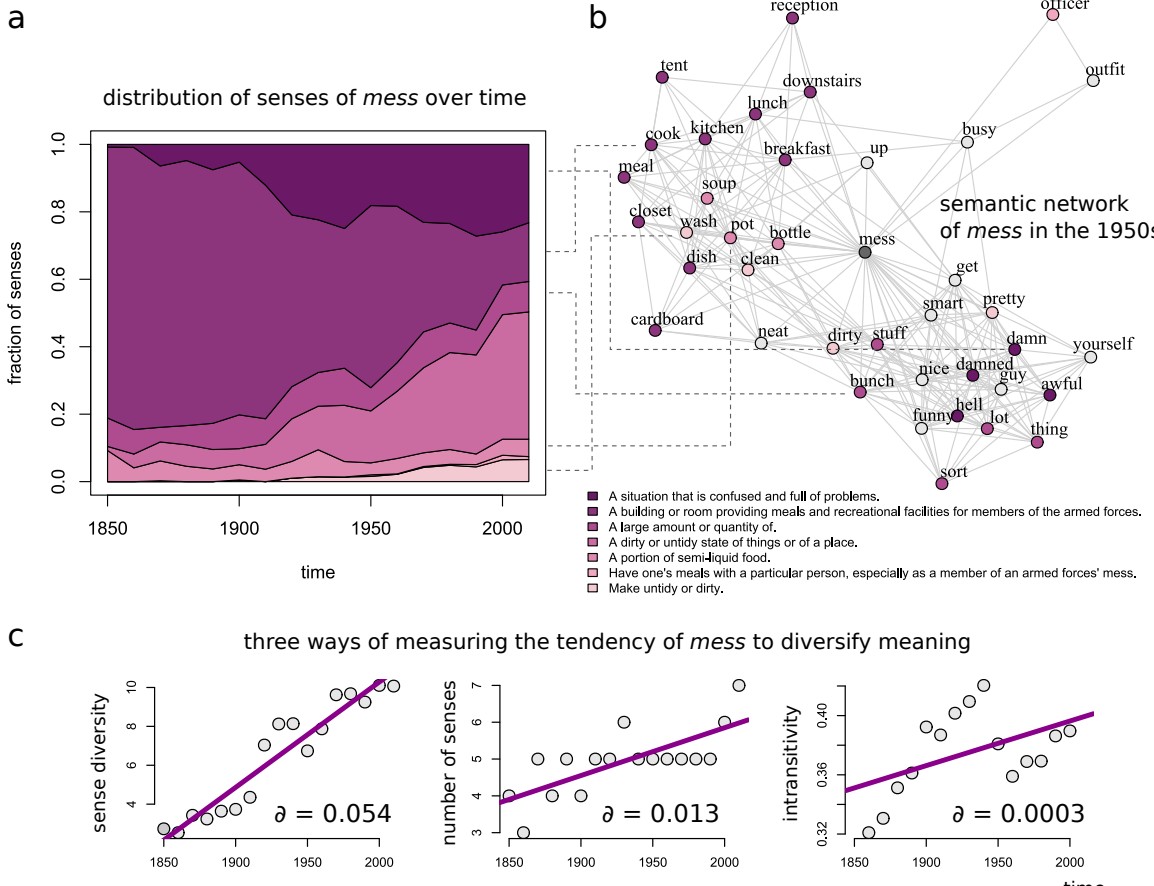

Figure 2: (a) The evolution of the probability distribution over all senses of the word *mess* from 1850 to 2000. In 1850, *mess* has had mainly the meaning 'room for meals'. The distribution of senses becomes more diverse over time. (b) Ego-network representing the semantic neighborhood of *mess* in 1950 (for this visualization, only cosine similarities above 0.25 are displayed as edges). At least two clusters can be identified. Nodes are color-coded with respect to their senses. (c) Developments of sense diversity, number of senses, and network intransitivity (see 5.1) of the word *mess* over time ($\partial$ denotes the respective slopes). All three developments indicate that *mess* became more polysemous.

## 5.1 Diversification tendency

The tendency to diversify meaning measures the extent to which a word becomes more polysemous over time. Crucially, this depends on how the degree of polysemy is measured for a given word at a given point in time. We operationalize the degree of polysemy in three ways.

First, we use the decade-wise probability distribution $(p_i)_{i=1,\dots,n}$ over all $n$ senses of a given word provided by Hu et al. (2019). A flat distribution indicates high diversity (i.e., high polysemy), while a distribution with, say, only a single dominant sense indicates low diversity. Hence, we derive an index of diversity given by $D = \exp(H)$, where $H = -\sum_{i=1}^{n} p_i \ln(p_i)$ is the Shannon entropy of the distribution (Hill, 1973). We refer to this as *sense diversity*.

Second, for a word in a given decade, we com-

pute the effective number of senses as the number $N$ of all senses with $p_i > 0.05$ (see Section 7 for discussion). That is, we ignore marginal senses when computing polysemy. We refer to this measure as the *number of senses*.

Third, we operationalize polysemy as in Hamilton et al. (2016) through the semantic neighborhood of a word. We base our analysis on the pre-trained SNGS embeddings used in that paper.[6] For each decade, we compute a network in which two words in our data set are connected if they are semantically close. We compute pairwise cosine similarity for all word pairs and add an edge whenever cosine similarity is greater than the median of all pairwise similarities in the data set. An example is shown in Figure 2b. For each word, we then compute its local clustering coefficient $C$ (i.e.,

[6] https://github.com/williamleif/histwords

transitivity), and define *intransitivity $I = 1 - C$*. As in Hamilton et al. (2016), we take intransitivity to measure polysemy (words tightly interconnected neighborhoods have few senses; words with star-like network topologies have many senses).

We compute sense diversity, the number of senses, and intransitivity for every word in every decade. For every word we fit three Gaussian linear regression models, one for each measure, i.e., $D(t) = b_D t + c_D + \varepsilon$, $N(t) = b_N t + c_N + \varepsilon$, and $I(t) = b_I t + c_I + \varepsilon$, where $t$ is time. We take the respective slopes $\partial_t D = b_D$, $\partial_t N = b_N$, and $\partial_t I = b_I$ to function as measures of the *tendency to diversify meaning* (see Figure 2c).

## 5.2 Non-conformism bias

A word's non-conformism bias measures to what extent that word is used in a non-conformist way. We study the usage of words across more and less conformist genres to obtain empirical estimates of non-conformism biases. Six different (sub-)corpora are used for this purpose. English Wikipedia[7], and sub-corpora from the Corpus of Contemporary American English (COCA)[8] covering magazines, news, and academic texts are selected as representing conformist genres, as they feature language obeying linguistic norms. The fiction sub-corpus of COCA, as well as a corpus of song lyrics[9], represent non-conformist genres in that the language therein is expected to be less constrained and more open to creative linguistic behavior.

For each word, we compute the mean per-million normalized frequency across all conformist corpora, $f_c$, and the mean per-million normalized frequency across all non-conformist corpora, $f_{nc}$. We define the strength of the *non-conformism bias* as $\beta = \log(f_{nc}/f_c)$, so that $\beta$ is positive if a word has relatively high frequencies in non-conformist genres as opposed to conformist genres.

## 5.3 Discriminability

Discriminability measures how well an individual can discriminate between two senses of a word. If discriminability is low, individuals perceive two

semantic variants of a word as similar. If discriminability is high, two semantic variants are more likely perceived as different senses of the word.

Discriminability cannot be easily measured based on corpus data, hence we rely on two different proxies in our analysis. We demonstrate in the appendix that both proxies correlate significantly with a measure that is closer to what one intuitively understands as semantic discriminability in a small set of words.

The first discriminability measure that we employ is lexical concreteness. Lexical concreteness is defined as the extent to which the content of a word can be grasped by the senses. It was shown that concreteness is subject to semantic prosody, i.e., concrete words are likely to surface in concrete contexts, and abstract words are likely to surface in abstract contexts (Snefjella and Kuperman, 2016). Under the assumption that this also extends to extra-linguistic contexts, we argue that concrete words (like *mess*) are more likely to be used in situations that let individuals differentiate between meanings than this holds true for abstract words (like *space*). We use crowd-sourced subjective ratings from Brysbaert et al. (2014) to measure concreteness.

The second discriminability measure is based on the sense descriptions from OED as provided by Hu et al. (2019) for all words. For each word, we computed embeddings using SentenceBERT[10] (Reimers and Gurevych, 2019) for all sense descriptions of that word, a standard approach to represent text. We then compute all pairwise correlations among all senses of the word (Pearson's $r$).[11] Discriminability is defined as the mean of all correlation coefficients. Thus, discriminability is low, if all senses are relatively close to each other, and high if, on average, senses differ from each other.

We find that the correlation between both measures (concreteness based vs. description-embedding based) is statistically robust at $r = 0.25$ ($CI_{0.95} = (0.22, 0.29)$). For additional support of the validity of these proxies see Appendix A.2.

## 5.4 Frequency

Two different ways to assess word frequency are used. For measuring frequency synchronically, we use contemporary frequency information from Brysbaert et al. (2014). For measuring frequency

---

[7] https://huggingface.co/datasets/wikipedia; sample of 70 million tokens.

[8] Word-frequency list from https://www.english-corpora.org/coca/; about 120 million tokens per genre.

[9] 200,000 English songs from https://genius.com/; 32 million tokens.

[10] https://huggingface.co/sentence-transformers/all-MiniLM-L6-v2

[11] Note that the correlation coefficient is equivalent with cosine similarity for centered data.

diachronically, we compute per million normalized token frequencies for each decade in COHA and average across all decades in the observation period. In both cases, frequency was log-transformed. Synchronic/contemporary and diachronic/historical frequencies correlate strongly and significantly with Pearson's $r = 0.76$ ($CI_{0.95} = (0.75, 0.77)$).

## 5.5 Statistical analysis and results

We need to measure the effects of (i) non-conformism bias $\beta$, (ii) discriminability $\delta$, and (iii) frequency $\varphi$ on the tendency to diversify meaning $T$. We have derived three measures for the dependent variable $T$ ($\partial_t D$, $\partial_t N$, and $\partial_t I$), two measures for bias $\beta$ (concreteness vs. description based) and two measures for frequency $\varphi$ (synchronic vs. diachronic), resulting in 12 different configurations of dependent and independent variables.

As slopes, estimates of the tendency to diversify meaning are not theoretically constrained to a particular interval, and none of the three measures of the independent variable shows a strongly skewed distribution. Hence, we fit Gaussian linear models of the form $T = b_\beta \beta + b_\delta \delta + b_\varphi \varphi + c + \varepsilon$ to our data. All variables were centered and normalized with respect to their standard deviation before entering the models in order to standardize regression coefficients $b_\beta$, $b_\delta$, and $b_\varphi$.

The estimated regression coefficients are shown in Table 2. It can be seen, first, that almost all effects turn out to be statistically significant, and, second, that all significant effects have the sign predicted by the theoretical analysis. That is, all configurations (i) effects of non-conformism biases are positive, (ii) effects of discriminability are positive, and (iii) effects of frequency are negative when predicting the tendency of diversification, and this holds true across all measures of the dependent variable. The one measure that stands out in this regard is $\partial_t I$, i.e., diversification tendency derived through lexical networks, which shows no statistically robust associations with non-conformism bias in two configurations. Most effects are within the weak to medium range (Cohen, 1992). Visualizations for all model configurations can be found in Appendix A.3.

## 6 Discussion and conclusion

In this paper, we have analyzed the evolution of lexical polysemy in two ways: first, by means of a mathematical model that captures the population

|  | $\partial_t D$ | SE | $\partial_t N$ | SE | $\partial_t I$ | SE |
|---|---|---|---|---|---|---|
| diachronic $\varphi$ and concreteness-based $\delta$ | | | | | | |
| $b_\beta$ | **0.06** | 0.02 | **0.06** | 0.02 | **0.07** | 0.01 |
| $b_\delta$ | **0.23** | 0.02 | **0.15** | 0.02 | **0.03** | 0.01 |
| $b_\varphi$ | **-0.10** | 0.02 | **-0.12** | 0.02 | **-0.20** | 0.01 |
| diachronic $\varphi$ and description-based $\delta$ | | | | | | |
| $b_\beta$ | **0.08** | 0.02 | **0.07** | 0.02 | -0.00 | 0.02 |
| $b_\delta$ | **0.08** | 0.02 | **0.04** | 0.02 | **0.06** | 0.02 |
| $b_\varphi$ | **-0.05** | 0.02 | **-0.09** | 0.02 | **-0.48** | 0.02 |
| synchronic $\varphi$ and concreteness-based $\delta$ | | | | | | |
| $b_\beta$ | **0.08** | 0.02 | **0.08** | 0.02 | -0.00 | 0.02 |
| $b_\delta$ | **0.08** | 0.02 | **0.06** | 0.02 | **0.12** | 0.02 |
| $b_\varphi$ | -0.02 | 0.02 | **-0.06** | 0.02 | **-0.22** | 0.02 |
| synchronic $\varphi$ and description-based $\delta$ | | | | | | |
| $b_\beta$ | **0.08** | 0.02 | **0.09** | 0.02 | **0.09** | 0.01 |
| $b_\delta$ | **0.23** | 0.02 | **0.16** | 0.02 | **0.03** | 0.01 |
| $b_\varphi$ | **-0.08** | 0.02 | **-0.10** | 0.02 | **-0.13** | 0.01 |

Table 2: Regression coefficients with standard errors for all models. Within each block, each column stands for one model configuration. Bold indicates statistically non-trivial effects at a 95% confidence level.

dynamics of word variants; second, through an empirical analysis covering the factors implemented in the population-dynamic model. The approaches yield converging outcomes.

As far as frequency is concerned, our results are in line with Hamilton et al.'s (2016) observation that frequency interacts with semantic change, namely in such a way that frequency in fact impedes change and fosters semantic stability. In a similar vein, Pagel et al. (2007) have shown that frequent concepts resist change more easily. Note that our way of conceptualizing semantic change subtly differs from that in Hamilton et al. (2016) who measured the impact of frequency on *any* type of change in word meaning. In contrast, the focus of our analysis is on a word's tendency to become more polysemous over time. Note, though, that our model not only predicts that low frequency entails, under certain circumstances, semantic diversification but also that high frequency leads to the existence of an attracting evolutionarily stable strategy in the semantic space, hence eventually demoting semantic change for high-frequency words altogether. That is, our model provides a mechanistic explanation for the empirical observation made in the literature: high-frequency words are more likely to adopt stable states in the semantic trait space that optimize ease of adoption.[12]

This explanation is an interesting alternative to

[12]Similarly, our model and the empirical account implicitly capture sense loss: multiple equilibria in the semantic trait space could lose their stable status if conditions (i-iii) in Proposition 1 were not fulfilled anymore.

the predominant account, which traces the positive relationship between frequency and semantic stability back to the mechanism of cognitive entrenchment (Bybee, 2006; Ellis et al., 2016). Here, the argument is that frequent words are strongly entrenched, i.e., subject to routinized perception, processing, and production so that they can resist change more easily (Schmid, 2020; Baumann et al., 2023).[13] Importantly, our account is agnostic with respect to mechanisms of entrenchment. In our model, the observed evolutionary behavior emerges from the assumption that frequent words are acquired more easily.

As a final note on the effect of frequency, it is worth mentioning that Hamilton et al.'s (2016) result was criticized as being a methodological artifact that arises from effects of frequency on cosine-similarities between word embeddings (Dubossarsky et al., 2017). While we do acknowledge this criticism it is important to emphasize that, from a methodological point of view, neither the theoretical nor the empirical account in our approach can be invalidated by these effects. Hence, our research provides independent evidence for Hamilton et al.'s (2016) findings.

In addition to the effect of frequency, our model predicts a positive relationship between non-conformism biases and the tendency of a word to establish more senses over time. The prediction is supported by our empirical account in ten out of twelve configurations. In the model, the mechanism that promotes the stable coexistence of two semantic senses is that of negative frequency dependence, i.e., an advantage of lexical variants if and as long as they are rare.

The prediction that this mechanism promotes polysemy is intuitive, and not very surprising for that matter. If rare variants have an advantage in their proliferation then this facilitates the establishment of innovative uses. This resembles the argument that is brought forth in the discussion about the role that extravagant usage plays in lexical change (Ungerer and Hartmann, 2020; Petré, 2017; Haspelmath, 2000).

We have also shown empirically and theoretically (in line with Baumann and Mühlenbernd (2022)) that non-conformism leads to diversifica-

tion of meaning only if discriminability is high. If discriminability is low then semantic innovations that are close to an established sense will be effectively perceived as coinciding with that sense. Consequently, they will suffer from the same disadvantage imposed by negative frequency dependence that all common types are subject to, even though these innovations are initially rare.

We consider it as a main strength of our approach that we do not only support (hypothesized) relationships between factors in semantic change and polysemy on *empirical* grounds, but that we can also point at the mechanisms driving semantic change based on our *theoretical* analysis. This is so, in particular, due to the two-level architecture of the mathematical model, which combines short-term population dynamics driven by linguistic interactions that are subject to assumed biases with evolutionary long-term dynamics of lexical semantics. In that sense, our work goes beyond correlational accounts of polysemy.

## 7 Limitations

Our approach is subject to multiple limitations, the most prominent of which are grounded in the abstract nature of our theoretical account. Our model is based on a set of simplifying assumptions. First, we assume that speakers in the population are homogeneously mixed (and no further population structure, e.g., with respect to age, gender, or social status). This is a standard assumption in many mathematical accounts to ecology (Hofbauer et al., 1998). In line with linguistic research (Hopper and Traugott, 2003) lexical change is assumed to be a gradual phenomenon, i.e., that innovative steps are relatively small. Furthermore, we assume innovations to spread fast, namely so fast that they reach their population-dynamic equilibrium (or vanish) before the next innovation is introduced. From a mathematical point of view, the latter two assumptions are necessary to model semantic change by means of an ODE in continuous time (i.e., equation (1)) (Dercole and Rinaldi, 2008).

Note, also, that we assume every speaker to use only one semantic variant at a time, according to our formulation of the model. That is, the stable coexistence of multiple senses, say A and B, in fact corresponds to the presence of two coexisting sub-populations, one using A and one using B. However, the model can easily be reinterpreted in such a way that the units in the population are not

---

[13]On the other hand, however, frequency was argued to drive semantic bleaching together with formal reduction (Hopper and Traugott, 2003), as evident in grammaticalization processes (e.g., the development of French *pas* from a noun to a negation particle).

individuals, but rather usage events of individuals.

Not all of the model's parameters and quantities can be measured empirically. Clearly, for example, there is no available historical data of the number of users $U$ of a word and its multiple senses. Likewise, we do not specify what the shape of adoption rate $\alpha$ looks like exactly. Importantly, however, we do not need to have this information for making predictions about the expected long-term evolution of the semantic property. This is because we are interested in qualitative assertions (does a branching point exist?) rather than quantitative ones (when did a certain word obtain a new sense?). The fact that we can derive *analytical* predictions is one of the main strengths of our approach.

Our empirical account is subject to limitations as well. First, many of the effects in the statistical models are relatively weak and sometimes display non-linearities that could be further examined (see Appendix A.3). Second, the corpus data might not be fully representative of the language spoken through the 19th and 20th century. Third, sense definitions in OED depend on decisions by the lexicographer. Note, also, that the word-sense identification model in Hu et al. (2019) has an accuracy of about 93.8%. This means that there is a non-negligible chance of the estimated proportions not being accurate. The threshold of 5% for assessing the number of senses is arbitrary, but it uniformly applies to all words and all periods.

Using concreteness ratings for our purpose, despite being motivated theoretically, can be problematic as they are based on subjective ratings (Brysbaert et al., 2014) and hence contemporary. This could be mitigated by considering historically reconstructed concreteness (Snefjella et al., 2019). Non-conformism estimates are contemporary as well. Follow-up research could examine historical estimates of non-conformism, albeit at the cost of a more limited range of genres than in our approach.

Sense descriptions in OED are often very short, so it is possible that their embeddings do not faithfully represent the respective senses. That said, we would like to point the reader to our robustness check in Appendix A.2 which renders our proxies of discriminability relatively reliable. Despite being subject to such limitations, we think that NLP methods are highly valuable for testing the predictions of analytical models employed in the study of diachronic linguistics.

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

# A  Appendix

## A.1  Proof of Proposition 1

In this section, we derive conditions under which equation (1) shows an evolutionary branching point (resulting in the stable coexistence of two semantic variants). Before that, we need to derive the population dynamics of single word variants given by (2) based on the assumptions (I-III) in Section 4. This is done in the following subsection.

### A.1.1  Population dynamics

In order to derive the equilibrium number of users of a word characterized by value $x$, i.e., $\hat{U}(x)$, we need to define the underlying population dynamics of the word. We will do so by setting out with a simple and well-studied model of lexical change that accounts for learning and frequency, and subsequently expand it to cover conformism and non-conformism biases.

First, we assume that for every individual word variant, the population of linguistic individuals can be split into users that know and use the word and non-users that do not use it. Let us denote the respective numbers of individuals as $U$ and $N$, respectively. Furthermore, we assume that populations are large and set $U + N = 1$. Non-users can adopt a word variant (like *mess* with a positive connotation) and switch to the user class at an adoption rate $\alpha$ whenever they encounter the word interacting with a user, which occurs at frequency $\varphi$. In

this simple scenario and under the assumption of homogeneous mixing of users and non-users, word dynamics are defined by the ordinary differential equation

$$\dot{U} = \varphi \, \alpha U N - U. \qquad (3)$$

This model is equivalent to previously studied models of cultural change (Cavalli-Sforza and Feldman, 1981) and lexical dynamics (Nowak, 2000; Nowak et al., 2000; Solé, 2011). This equation models how words spread through a speaker population in an S-shaped manner; a pattern often encountered in diachronic linguistics (Denison, 2003). Note that, apart from variants of the same word (such as positive and negative variants of *mess*), we do not model any interactions between different word types (like hypothetical co-developments and interactions of *mess* and *banana*).

We assume that $\alpha$ is a real-valued smooth function of the value of the semantic property $x$ and that there is a value $x_0$ for which $\alpha(x)$ obtains a maximum within $\mathbb{X}$. Sentiment, for instance, was shown to influence lexical processing, which in turn arguably impacts a word's adoption rate. Note, importantly, that $\alpha$ does not only cover word acquisition by children but also adoption of words and their variants by adults.

Let us integrate conformism and non-conformism rates into the model. As discussed above, conformism rates represent effects of positive frequency dependence *in addition* to frequency dependent word adoption. When two users meet, we assume that word usage is strengthened at a *conformism rate* $\kappa$. Similarly, if a user of a word meets another user of that word, they will abandon that word (or word variant) due at a *non-conformism rate* $\nu$. In that case they switch back to the non-user class.

It is plausible that, whenever two users of slightly different variants $x$ and $y$ in $\mathbb{X}$ of the word meet, conformism and non-conformism rates depend on the difference between those two variants. This is because if both variants are very dissimilar then these two items are not so likely to be seen as two variants of the same word in the first place. In that case, conformism and non-conformism biases are expected to vanish. We model this behavior by letting rates $\kappa$ and $\nu$ depend on the difference $\Delta = x - y$ in a Gaussian manner as in

$$\kappa(\Delta) = \kappa_0 \exp(-1/2 \cdot \Delta^2 \delta^2) \qquad (4)$$

and

$$\nu(\Delta) = \nu_0 \exp(-1/2 \cdot \Delta^2 \delta^2), \qquad (5)$$

with variance $1/\delta$, so that (non-)conformism rates are highest if $x$ and $y$ coincide (obtaining the maxima $\kappa_0$ and $\nu_0$, respectively). Let us define *non-conformism bias* as $\beta := \nu_0 - \kappa_0$. The Gaussian dependency of (non-)conformism rate on $\Delta$ is motivated by the assumption that individuals are influenced to a lesser extent by other individuals that display substantially different linguistic behavior (Dercole and Rinaldi, 2008; Kim and Hommel, 2015).[14] For instance, non-conformism rates (i.e., the need to abandon one's own behavior) is assumed to be highest if the linguistic behaviors of two individuals meeting each other coincide.

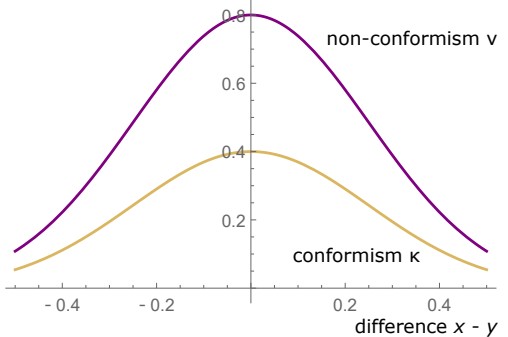

Figure 3: Conformism rate $\kappa(\Delta)$ and non-conformism rate $\nu(\Delta)$, both with $\delta = 4$. Maxima are at $\kappa_0 = 0.4$ and $\nu_0 = 0.8$, so that the curvature of $\nu$ is higher than that of $\kappa$ at $\Delta = 0$.

In both functions, $\delta$ can be interpreted to measure the users' ability to discriminate between usage variants within $\mathbb{X}$. I.e., $\delta$ measures the users' *discriminability*. We assume, for simplicity, that for any given word, $\delta$ is constant across the whole speaker population. If individuals are equipped with fine-grained semantic perception (high $\delta$), they can easily distinguish between, say, positive and negative variants of the word *mess*. In that case, (non-)conformism rates are reduced faster as $\Delta$ increases. If, in contrast, individuals can not so easily distinguish between different usage variants (low $\delta$), they are more likely to treat these variants as the very same item which in turn yields high (non-)conformism rates. See Figure 3.

In total, for a single variant characterized by $x$,

---

[14]Similar functions were used to model perceptual similarity (Nosofsky, 1986; Jäger, 2006) and communicative vagueness (Franke and Correia, 2018)

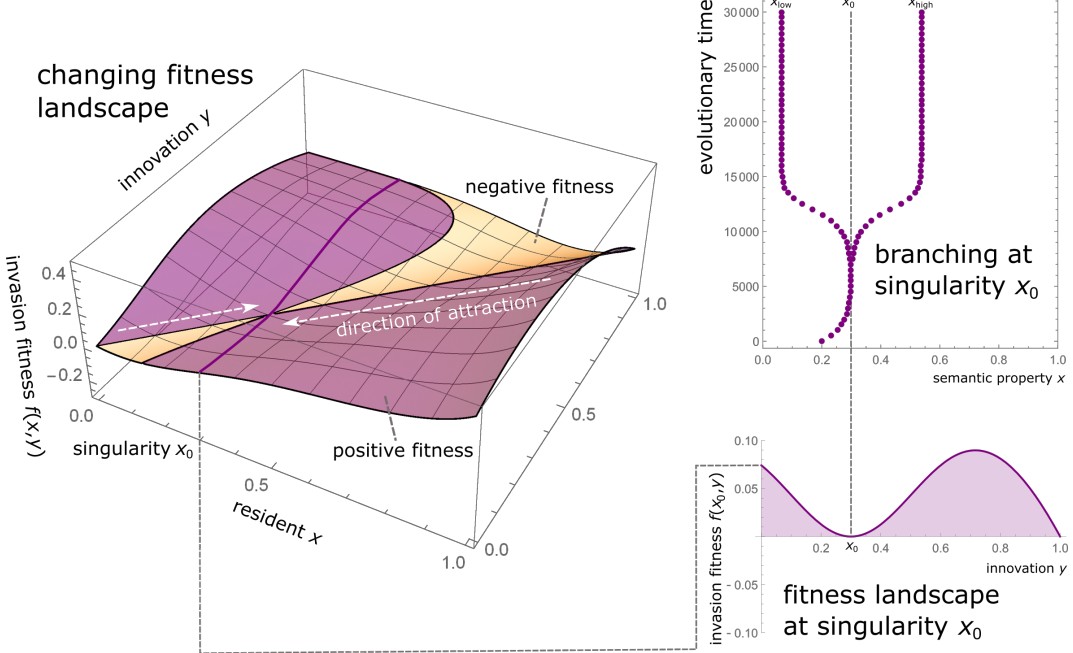

Figure 4: Evolutionary dynamics of $x$ in the semantic subspace $\mathbb{X}$ undergoing branching. In this setup, the adoption rate is defined as a concave function $\alpha(x) = -(x - x_0)^2 + \alpha$ with $x_0 = 0.3$ and $\alpha = 0.5$. Discriminability is set to $\delta = 4$, and frequency is $\varphi = 1$. Left: changing fitness landscape defined by invasion fitness $f(x, y)$. Every resident $x$ defines a different fitness landscape that innovations $y$ must cope with. Purple (dark) denotes areas where innovations can invade (positive $f$) the resident variant. Through multiple invasion-substitution events, semantic property $x$ approaches the singularity $x_0$ (direction of evolution shown by white arrows). This is because for every resident, innovations closer to $x_0$ have positive invasion fitness $f$ (purple area). Bottom right: fitness landscape defined by singularity $x_0$ (i.e., the slice represented by the dark line in the changing fitness landscape). Here, $x_0$ is a local minimum of its own landscape so that it can be invaded by innovations on both sides. Top right: evolutionary trajectory of $x$ given equation (1). Property $x$ first approaches $x_0$, where it splits into two variants that converge towards stably coexisting variants $x_{high}$ and $x_{low}$, i.e., an instance of evolutionary branching.

the population dynamic systems then reads

$$\dot{U} = \varphi\,\alpha(x)UN + \kappa(\Delta = 0)UU - \nu(\Delta = 0)UU. \tag{6}$$

This ODE is an extension of the model of religious change in Doebeli and Ispolatov (2010b) and that of lexical change in Baumann and Mühlenbernd (2022) in that it features both, conformism *and* non-conformism effects, as well as a parameter measuring frequency (as does the model by Nowak et al. (2000); see (3) above). It does also not feature a term that logistically regulates population growth, in contrast to the model in Doebeli and Ispolatov (2010b).

The system above shows a non-trivial equilibrium at $\hat{U}(x) = \varphi\,\alpha(x)/(\varphi\,\alpha(x) - \kappa_0 + \nu_0) > 0$. It exists and is stable as long as $\alpha(x) > 0$ and $\varphi\,\alpha(x) + \nu_0 > \kappa_0$, which can be seen by linearizing around $\hat{U}(x)$ in (2) (Hofbauer et al., 1998). That is, words must be adoptable and conformism rates must not be too strong.

### A.1.2 Invasion fitness

We still need to derive invasion fitness $f(x, y)$ in equation (1). Invasion fitness is defined as the growth rate of an initially rare innovation characterized by $y$ in a population of speakers that predominantly use $x$ (i.e., the resident population) (Metz et al., 1995; Dercole and Rinaldi, 2008; Doebeli, 2011). That is, the number of users of the variant with value $y$ is assumed to be close to zero and the resident population characterized by value $x$ rests at its population dynamic equilibrium $\hat{U}(x)$. That is, in the beginning $y$ users only interact with $x$ users but not with other $y$ users. Hence,

$$f(x, y) = \varphi\,\alpha(y)(1 - \hat{U}(x)) + \kappa(y - x)\hat{U}(x) - \nu(y - x)\hat{U}(x). \tag{7}$$

For every resident variant $x$, invasion fitness $f(x, y)$ defines a fitness landscape that users of innovative variants characterized by value $y$ must cope with. If $f(x, y) > 0$ then an innovation characterized by

$y$ can successfully invade the resident population characterized by $x$. Furthermore, if $y$ is not close to an evolutionary singularity (see below), the innovation will, if it successfully invades, also replace the resident. Thus, invasion fitness determines the direction of the evolutionary dynamics of $x$. More specifically, evolutionary progress is determined by the *fitness gradient* $D(x) = \frac{\partial f(x,y)}{\partial y}\Big|_{y=x}$ in (1). An *evolutionary singularity* then is a point $\hat{x} \in \mathbb{X}$ where $D(\hat{x}) = 0$, i.e., an equilibrium of equation (1). That is, invasion fitness lets us identify the long-term evolutionary dynamics of semantic property $x$.

### A.1.3 Conditions for diversification

Let us now study the long-term dynamics of equation (1) given the quantities derived in the previous sections. We have seen that for identifying evolutionary singularities in this equation it is sufficient to analyze the fitness gradient $D(x)$. Since $\kappa$ and $\nu$ obtain maxima if $x = y$, we have that $D(x) = \varphi \alpha'(x)$. By assumption, adoption rate $\alpha$ obtains a maximum at $x_0$ (see definition of $\alpha$ above). This entails, first, that $x_0$ is an evolutionary singularity and, second, that the dynamics of $x$ given by equation (1) approach $x_0$. Hence, $x_0$ is an *evolutionary attractor*.

The question now is this: what happens if the semantic property $x$ comes close to $x_0$? For this, we need to analyze the fitness landscape $f(x_0, y)$ defined by $x_0$ that nearby innovations $y$ are exposed to. If $x_0$ is a maximum of this fitness landscape then nearby innovations cannot invade. In this case, $x_0$ is *evolutionarily stable*. If, however, $x_0$ is a minimum of this fitness landscape, then nearby innovations on both sides of $x_0$ can invade and co-exist. In that case, $x_0$ is an *evolutionary branching point* (Geritz et al., 1997). It is this evolutionary scenario which leads to diversification of semantic properties and hence to polysemy.

Whether $x_0$ is a minimum of this fitness landscape is given by the local curvature of $f(x_0, y)$, that is,

$$
\begin{aligned}
T_{x_0} &:= \frac{\partial^2 f(x_0, y)}{\partial y^2}\bigg|_{y=x_0} \\
&= \varphi\, \alpha''(x_0)(1 - \hat{U}(x_0)) \\
&\quad + (\kappa''(0) - \nu''(0))\hat{U}(x_0).
\end{aligned} \tag{8}
$$

If $T_{x_0} > 0$ then $x_0$ is a minimum of $f(x_0, y)$ (and a maximum if $T_{x_0} < 0$). In other words, we can

interpret $T_{x_0}$ as a measure of the *tendency to diversify meaning*. In particular, we see that $T_{x_0}$ depends on a word's frequency $\varphi$, non-conformism bias $\beta = \nu_0 - \kappa_0$, and discriminability $\delta$. By taking the derivatives of $T_{x_0}$ with respect to $\varphi$, $\beta$, and $\delta$, we see that

$$
\frac{\partial T_{x_0}(\varphi, \beta, \delta)}{\partial \varphi} = \underbrace{\alpha''(x_0)}_{<0}(1 - \hat{U}(x_0)) < 0, \quad (9)
$$

because $\alpha$ is locally concave around $x_0$ by assumption,

$$
\frac{\partial T_{x_0}(\varphi, \beta, \delta)}{\partial \beta} = -\underbrace{\frac{\partial e^{-\frac{\Delta^2 \delta^2}{2}}}{\partial^2 \Delta}\bigg|_{\Delta=0}}_{<0} \hat{U}(x_0) > 0,
$$

$$
\tag{10}
$$

since $\exp(-1/2 \cdot \Delta^2 \delta^2)$ is concave at its maximum, and

$$
\begin{aligned}
\frac{\partial T_{x_0}(\varphi, \beta, \delta)}{\partial \delta} &= -\beta \frac{\partial}{\partial \delta}\left[\frac{\partial e^{-\frac{\Delta^2 \delta^2}{2}}}{\partial^2 \Delta}\bigg|_{\Delta=0}\right]\hat{U}(x_0) \\
&= \underbrace{-\beta \frac{\partial}{\partial \delta}\left[-\delta^2\right]}_{>0}\hat{U}(x_0) > 0,
\end{aligned}
$$

$$
\tag{11}
$$

as long as $\beta = \nu_0 - \kappa_0 > 0$.

Hence, an evolutionary branching point exists if $\varphi$ is sufficiently low, and if $\beta$ and $\delta$ are sufficiently high.

### A.2 Robustness check for discriminability

In this section, we test to what extent both measures of discriminability (concreteness based; sense-description based) correlate with more direct estimates of discriminability. For this, we use the WiC data set (Pilehvar and Camacho-Collados, 2019), which contains for a set of target words several sentence pairs together with ratings indicating whether the target word has the same sense in both sentences or not (1 for *same sense* vs. 0 for *different sense*). We used the whole data set (train, dev, test).

For each target word, we computed the mean rating (*mean sameness*) of all sentence pairs for that word, thereby only considering words with 20 or more sentence pairs to avoid inaccurate mean estimates. This leaves us with 50 different target words that also show up in our set of 3165 words. Arguably, a high mean sameness rating indicates that different contextual uses of a single word are

not easy to discriminate, while a low mean sameness rating indicates high discriminability. Consequently, we would expect mean sameness to negatively correlate with both of our discriminability proxies.

The correlogram in Figure 5 shows that both discriminability proxies correlate negatively with mean sameness as expected. While this supports our assumption that concreteness and average correlation among sense-description embeddings function as good proxies for discriminability, we do need to emphasize that this robustness check is based on a relatively small number of words all of which show high utterance frequency. Future research would need to consider a larger and more representative set of words.

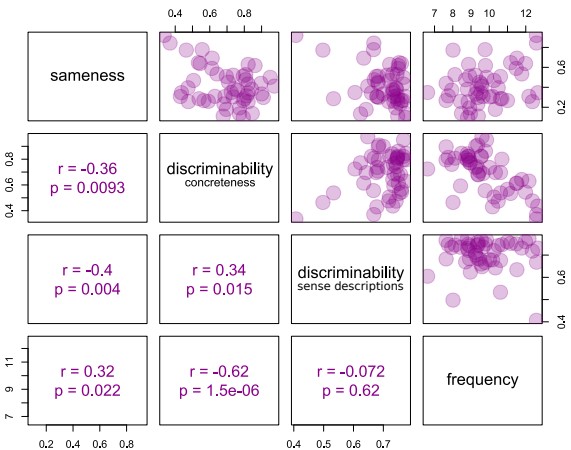

Figure 5: Correlogram of mean sameness, frequency, and both discriminability proxies (Pearson's $r$ together with p-values).

## A.3 Visualization of statistical models

Figures 6 and 7 show the effects of $\nu$, $\delta$, and $\varphi$ on the tendency to diversify meaning (three different measures) for all twelve configurations (rows). The shaded areas represent 95% confidence boundaries. Each point (gray) represents a single word.

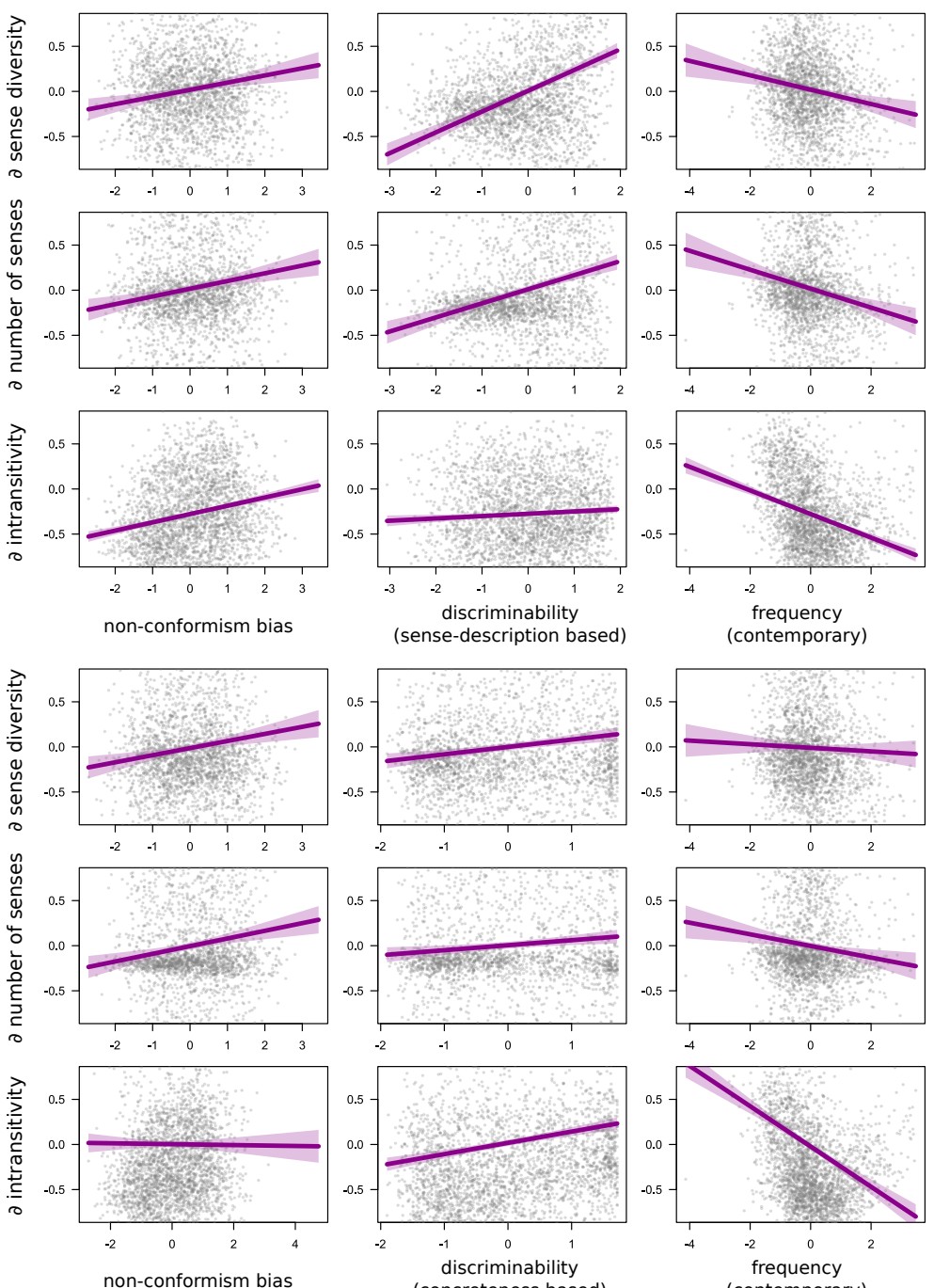

Figure 6: Model configurations employing contemporary word frequency.

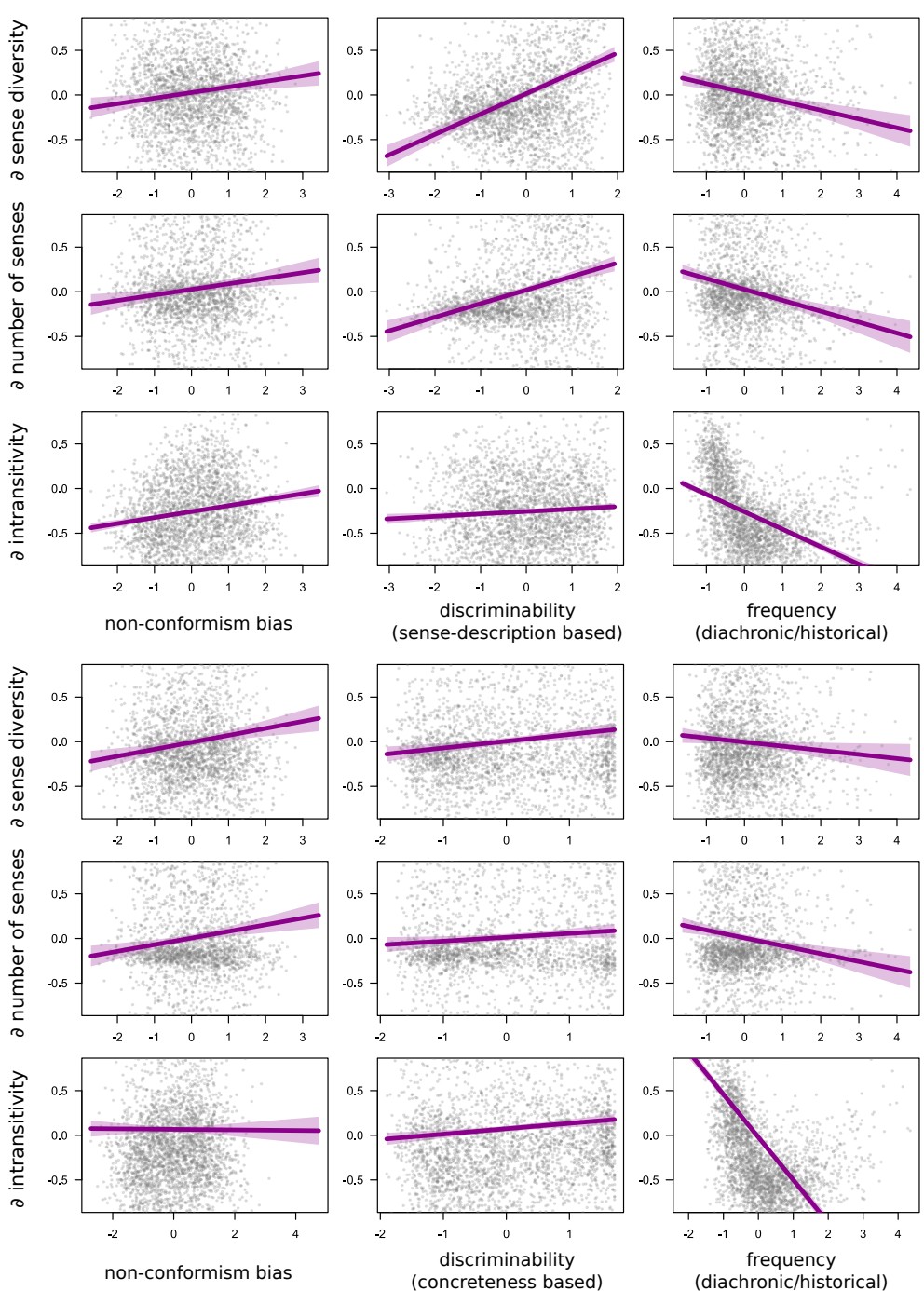

Figure 7: Model configurations employing diachronic/historical word frequency.