# OpenReview forum: "Seeing through the mess: evolutionary dynamics of lexical polysemy"
_EMNLP/2023/Conference — EMNLP 2023 Main_

### Official Review · Reviewer_gMYd · 2023-08-01

**Soundness:** 5

**Excitement:**

4: Strong: This paper deepens the understanding of some phenomenon or lowers the barriers to an existing research direction.

**Paper Topic And Main Contributions:**

The paper studies under what conditions a branching point in semantic space leading to stable polysemy is expected to happen, considering three factors shown to affect long-term semantic change, i.e., frequency, discriminability, and non-conformist usage of lexical items. Through the use of population dynamics models, the authors present a theoretical and empirical analysis of the proposed model, highlighting how the presence of a branching point relates to low frequency, non-conformist usage and high discriminability, i.e., words are likelier to acquire a novel stable sense if they are used rarely, if users have non-conformist biases (thus prefer to use words in novel ways), and if usages of a same lexical item have higher discriminability. Frequency, discriminability, and polysemy on the evolutionary scale are measured in different ways to ensure patterns are robust. The empirical analysis relies on a curated set of a few thousand words with extensive sense annotations, as well as on several secondary data sources including historical and synchronic corpora (COHA and COCA), and lexica (cocnreteness and frequency ratings).

The main contribution is certainly in the adoption of models of population dynamics used in biology and economics in the context of semantic change, with an outstanding mathematical rigor and the combination of a theoretical analysis with an empirical analysis.

Overall I liked the paper: there clearly is a lot of work to do in quantifying all variables that enter such complex population dynamics, but works such as this hold promise in giving the community tools to bring this work forward in principled ways and I think this paper should make it into the main conference.

----------------------------------------------------------

I thank the authors for their answer: I haven't eventually hanged my scores as I don't think the paper qualifies as transformative. That said, I do find it interesting and methodologically sound.

**Questions For The Authors:**

at page 2 you mention that none of the assumptions _immediately_ relates to semantic long term evolution. What do you mean by immediately? I agree they model short-term interactions, but it's hard for me to think of a scenario where long-term change does not depend on short-term interactions. Could you add a footnote outlining one such scenario that would make the non-trivial link more explicit?

at page 5 you mention that early subcorpora from the coha are relatively small: what is relatively? would your analysis change if you include those decades as well?

page 7, top: you mention corpus of historical american english but abbreviate it as coca, which is which? from context I guess you use the coca here, but not sure.

You interpret a correlation of 0.25 as reasonably strong: I agree that it's nothing worrying in the context of this work, considering the degree of noise involved in quantifying such complex phenomena, but that correlation is hardly strong: could you clarify what leads you to dub it as such?

page 8, top: what is t? the previous formula includes T, but not t. Maybe I'm missing something obvious, but I'm missing it nonetheless.

**Reasons To Accept:**

The paper is thorough, upfront about the assumptions it needs to make, why they need to be made, and what consequences they may have on results. The mathematical specifications are dense but overall clear also to a non-specialist like me. The results are interesting and contribute to present the methodology as an interesting tool to study diachronic phenomena in language, potentially offering other researchers an interesting springboard for future studies. The limitations section is very thorough and insightful, complementing the results presented in the main paper in important ways.

**Reasons To Reject:**

The paper is very dense in its presentation of the methodology and goes over the results rather quickly. The discussion is also not particularly insightful, and does not offer a very strong explanation for the tension between the observation that frequency is positively correlated with polysemy under a synchronic perspective and the empirical finding that sense differentiation tends to be stronger for low frequency words. The account offered by the authors (low frequency words from the past may become more frequent because of sense differentiation) would suggest that over time different word types should be frequent, but the correlation between frequency estimates over time seems to contradict this. While I don't think a surprising result should justify rejection, I think more care is warranted in interpreting these results than shown in the discussion.

**Reproducibility:**

4: Could mostly reproduce the results, but there may be some variation because of sample variance or minor variations in their interpretation of the protocol or method.

**Reviewer Confidence:**

4: Quite sure. I tried to check the important points carefully. It's unlikely, though conceivable, that I missed something that should affect my ratings.

**Typos Grammar Style And Presentation Improvements:**

Overall clearly written, in spite of how dense the mathematical treatment is, kudos! I signalled potential typos in the questions to the authors because their solution isn't obvious to me.

---

> ### Author Rebuttal · Authors · 2023-08-25
>
> We want to express our gratitude to the reviewer for their thorough, detailed, and reassuring feedback. We are excited to read that the reviewer considers our work as “springboard for future studies” and we are glad that the reviewer liked the way in which we presented our analysis.
>
> The main criticism is that the discussion section deserves a more in depth assessment of our results. We agree and we will gladly expand that section. In particular, the reviewer has addressed our account of the mismatch between the obtained results regarding the role of frequency (essentially confirming Hamilton et al. 2016) and Zipf’s meaning-frequency law. It seems that our discussion in this regard was more prominent than we intended it to be. In fact, the suggested causal chain (more senses lead to higher frequency) is rather a conjecture than something that can be derived directly from our theoretical or empirical analysis (this is, because frequency is an input parameter in the model and not the outcome). Indeed, to address said conjecture one would need to parametrize *growth* in frequency. This would entail an entirely different analysis, and we will make clear in the revision that answering this question is subject to future research.
>
> What we will do, anyway, is to expand the Discussion by carefully addressing the role of the parameters as integrated in the model. More specifically, we will move the discussion of frequency that is currently in the Limitations section (lines 690-696) to the Discussion section. Here, we show why our model predicts high-frequency words to be less prone to diversification: high-frequency words are more likely to adopt stable states in the semantic space. We will relate this (as we think interesting) observation to extant research (e.g., Pagel et al. 2007, *Nature*, 449).
>
> Other comments:
> * page 2, “immediately”: What we mean is that these factors influence long-term evolution only indirectly via multiple small-scale interactions (just as the reviewer correctly writes). We will rephrase this to make our point clear.
> * page 5, “relatively small” subcorpora: We need to be specify more clearly. In addition to a smaller corpus size (e.g., the sub-corpus of the first decade is only about half as big as that of the 1850s), the first decades of the 19th century also lack a newspaper subcorpus, hence early sub-corpora are less representative. That said, to be on the safe side we now ran the analysis with the whole time span and did not find any substantial differences in our results. We will put this information into a footnote.
> * page 7, “reasonably strong”: Yes, good point, we will change that to “medium strength” (as suggested by Cohen 1992). We actually wanted to express that it is “not weak”, but “medium” is more accurate.
> * page 8, “where *t* is time”: This was an editing error (actually, this phrase belongs to the model formulas in the end of section 5.1). Thanks for spotting this!

---

### Official Review · Reviewer_SeWs · 2023-08-04

**Soundness:** 2

**Excitement:**

2: Mediocre: This paper makes marginal contributions (vs non-contemporaneous work), so I would rather not see it in the conference.

**Paper Topic And Main Contributions:**

This submission presents a model attempting to predict the growth of polysemy for individual word forms diachronically. Why or how do some words gain many additional senses throughout the lifetime while others may not. Two big issues limit the effectiveness of the work however.

Firstly, the real theoretical status of polysemy is potentially in doubt; at the very least there is a fuzzy boundary between polysemy and homophony. When sound change causes new homophones to occur in a language, some of those will inevitably have similar meanings while others won't -- do we want to count those cases as one or the other, or rather distinguish accidental from intentional (extended) polysemy? How about when words lose senses over time?

Second, the issue in my opinion is whether this question falls into a class of problems with so many potential answers that it may prove impossible to distinguish them. i.e. there are so many potential underlying mechanisms which lead to polysemy that the mere existence of polysemy is not something that is capable of distinguishing theories of language development and change -- let alone support unique explanations on its own right.

Finally, despite the authors claims I don't see how The model mathematically combines a number of factors suggested to correlate with the growth of polysemy into a single implementation but none of these form a causal whole, so the work does not seem fundamentally different from a kind of correlational analysis.

**Reasons To Accept:**

The paper is generally well-structured and the topic (while not traditionally in the purview of NLP) may be of interest to audience at the conference.

**Reasons To Reject:**

As noted above I have concerns about the status of the phenomenon (polysemy) being modeled. About how distinguishable different causes of polysemy really are from one another, and whether the present work is really separable from the kind of correlational analysis they mention.

**Reproducibility:**

3: Could reproduce the results with some difficulty. The settings of parameters are underspecified or subjectively determined; the training/evaluation data are not widely available.

**Reviewer Confidence:**

4: Quite sure. I tried to check the important points carefully. It's unlikely, though conceivable, that I missed something that should affect my ratings.

---

> ### Author Rebuttal · Authors · 2023-08-25
>
> We would like to thank the reviewer for their feedback and time invested into carefully going through the manuscript. The review addresses the following three points:
>
> **(1) Polysemy and homonymy/homophony:**
> This is an important point and we will delimit the phenomenon under investigation (polysemy) from homonymy in the revision.
> We follow the mainstream distinction between polysemy and homonymy in linguistics. In short, the phenomenon that we look at is polysemy, i.e., one-to-many form-meaning mappings that come about via changes on the meaning side. The key criterion is that polysemy involves historical or psychological relatedness (cf. Leech, 1974, *Semantics*. Penguin Books; McMahon, 1994, *Understanding language change*. CUP). This is exactly what is modeled in our theoretical account (innovative steps in a semantic dimension originating from one variant).
>
> In contrast, homonymy (a rarer phenomenon) comes about via changes on the form side. Note that the historical information that our empirical analysis uses (Hu et al. 2019) excludes homonyms as listed in the OED as well so that the theoretical and the empirical account are consistent, and we will point this out more clearly in the final version of the paper.
> Note also that major sound changes leading to homophony (e.g., *steak* vs. *stake*) took place in the transition from the Middle to the Early Modern English period, ca. 15th century, and the data used in our empirical analysis stemming from after 1850 is relatively stable with regard to large-scale sound changes that could lead to new homophones.
>
> As to sense loss: yes, this is a relevant phenomenon that is explicitly treated in the empirical part and implicitly in the theoretical model (shifts in the model parameters can render multiple equilibria of eqn (1) unstable as per eqn (8)). We will add this point to the discussion section.
>
> **(2) Distinguishable causes of polysemy:**
> We agree that polysemy can have different causes and our research provides an account for how potential causes can be linked via a plausible population dynamic model, and we show that the predictions of the model are in agreement with empirical data. The main objective of our research is exactly to investigate a selection of them that is theoretically and empirically motivated.
>
> We are not aware of other research that attempts to mathematically model dynamics of polysemy linking individual behavior with phenomena on the population level, *and* for which results are tested against empirical observations. This does not mean that no alternative theory could account for the observed phenomena in principle, but it would not be adequate to characterize our approach as arbitrary.
>
> Our approach is not to derive “unique explanations on its own right” as the reviewer puts it. Quite to the contrary: our model shows that polysemy results from a well-motivated *combination* of different factors (e.g., non-conformism promotes diversification but only if discriminability is high enough).
>
> **(3) Causal vs. correlational analysis:**
> Our theoretical (mathematical) account allows for a causal assessment in that we start with mechanisms that apply to short-term interactions of speakers and rigorously analyze the evolutionary long-term consequences that follow from said mechanisms. The mathematical model (in fact, two stacked models) is far from representing a correlational analysis. Rather, we test its implications by showing that factors correlate as predicted.
>
> The reviewer argues that the “model mathematically combines a number of factors suggested to correlate with the growth of polysemy”. However, this is not exactly what we do. In fact, the first model combines factors that have an impact on linguistic *interactions*, and only in a subsequent step we check, in a second model, what their impact on growth in polysemy is. Crucially, the first model integrates the factors in a plausible way (Table 1). Extant (statistical) correlational studies, in contrast, *directly* relate factors to (dynamics in) polysemy. The strength of our account, we think, is that it adds a theory to these relationships. We will elaborate on these points in the revised version to make this clear.

---

### Official Review · Reviewer_VVLs · 2023-08-05

**Soundness:** 4

**Excitement:**

3: Ambivalent: It has merits (e.g., it reports state-of-the-art results, the idea is nice), but there are key weaknesses (e.g., it describes incremental work), and it can significantly benefit from another round of revision. However, I won't object to accepting it if my co-reviewers champion it.

**Missing References:**

In the list there are examples from the most relevant references on the topics.

**Paper Topic And Main Contributions:**

The paper proposes a formalisation, and related data analysis, of the diachronic phenomenon of lexical polysemy.
The noun “mess” is taken as example to show how the lexical change is (inversely) related to frequency (i.e. word usage by speakers), non-conformism, which in turn is related to high discriminability.

**Reasons To Accept:**

The theoretical- formal - approach is very clearly defined, and the topics (diachronic lexical change and polysemy) are important for a better understanding of language structure and use.
Results are very clearly and convincingly presented.

**Reasons To Reject:**

I am not completely sure that a formal approach, rather than a simulative one, would be of interest for the nlp community.

**Reproducibility:**

3: Could reproduce the results with some difficulty. The settings of parameters are underspecified or subjectively determined; the training/evaluation data are not widely available.

**Reviewer Confidence:**

4: Quite sure. I tried to check the important points carefully. It's unlikely, though conceivable, that I missed something that should affect my ratings.

---

> ### Author Rebuttal · Authors · 2023-08-25
>
> We would like to thank the reviewer for their positive feedback. It is good to see that our approach was presented in a clear way.
>
> It seems that the only substantial criticism is that we have opted for an analytical (mathematical) modeling approach over one that employs simulations, i.e., in the present context, agent-based simulation models. We do acknowledge the usefulness of simulation models in the study of language change. In particular, we see that, as argued by Steels (2016, *Philos. Trans. R. Soc. B*, 371), agent-based accounts allow for a causal assessment of mechanisms that drive language change (as do mathematical models like the ones employed here), and it is evident that simulation models can integrate a larger number of parameters and factors than is possible for mathematical models.
>
> The clear advantage of mathematical models, however, is that they let us examine relationships among parameters and outcomes analytically, i.e., by means of a mathematical expression, rather than statistically. We argue that this has benefits for the study of language, too (clearly, ours is by far not the first account to study language change with the help of mathematical models, let alone the study of biological evolution): our theoretical analysis yields deterministic and, crucially, testable predictions and we do not need to bother about issues like sample size, number and duration of simulation runs. We will elaborate on these advantages in the discussion section of the revised manuscript.

---

### Meta-Review · Area_Chair_LJwt · 2023-09-19

**Recommendation:** 3

**Metareview:**

The reviewers agree on the fact that this paper provides a valuable contribution to investigate the diachronic phenomenon of lexical polysemy, by providing a clearly defined theoretical and formal approach to gain a better understanding of language structure and use.
The main concern is related to the relevance of this formal approach for the nlp community. The methodology presentation is dense and would necessitate further details, in particular concerning the interpretation of results.

---

### Decision · Program_Chairs · 2023-10-07

**Decision:**

Accept-Main

**Comment:**

The reviewers agree on the fact that this paper provides a valuable contribution to investigate the diachronic phenomenon of lexical polysemy, by providing a clearly defined theoretical and formal approach to gain a better understanding of language structure and use.
The main concern is related to the relevance of this formal approach for the nlp community. The methodology presentation is dense and would necessitate further details, in particular concerning the interpretation of results.